# Virtual MOS Sensor Array Design for Ammonia Monitoring in Pig Barns

**DOI:** 10.3390/s25082617

**Published:** 2025-04-20

**Authors:** Raphael Parsiegel, Miguel Budag Becker, Pieter Try, Marion Gebhard

**Affiliations:** Group of Sensors and Actuators, Department of Electrical Engineering and Applied Sciences, Westphalian University of Applied Sciences, 45897 Gelsenkirchen, Germany; raphael.parsiegel@studmail.w-hs.de (R.P.);

**Keywords:** livestock monitoring, MOS sensor, ammonia monitoring, electronic nose, temperature-cycled operation, virtual sensor array, smart sensor, smart farming, machine learning

## Abstract

Animal welfare in barns is strongly influenced by air quality, with gaseous emissions like ammonia posing significant respiratory health risks. However, current state-of-the-art ammonia monitoring systems are labor-intensive and expensive. Metal Oxide Semiconductor (MOS) sensors offer a promising alternative due to their compatibility with sensor networks, enabling high-resolution ammonia monitoring across spatial and temporal scales. While MOS sensors exhibit high sensitivity to various volatile compounds, temperature-cycled operation is commonly employed to enhance selectivity, effectively creating virtual sensor arrays. This study aims to improve ammonia detection by designing a virtual sensor array through a cyclic data-driven approach, integrating machine learning with solid-state sensor modeling. The results of a two-week dataset with measurements of four different pig barns demonstrate ammonia sensing with a sampling rate of about 2/min and a range of 1–30 ppm. The method is robust and exhibits a 10% increase in normalized RMSE when comparing testing results of an unseen sensor module with results of the training dataset. A filter membrane boosts accuracy and prevents data loss due to contamination, such as flyspecks. Overall, the used MOS sensor BME688 is effective and economical for widespread continuous ammonia monitoring and localization of ammonia sources in pig barns.

## 1. Introduction

Ammonia (NH_3_), as a form of reactive nitrogen, represents a significant environmental concern, as it leads to eutrophication, acidification, and finally to a loss of biodiversity in ecosystems [1]. NH_3_ is also responsible for the formation of secondary fine particulate matter that causes health hazards such as increased respiratory and cardiovascular morbidity and mortality [2]. According to the European Environment Agency, between 1990 and 2021, agriculture emitted 94% of atmospheric ammonia. In addition to the application of synthetic nitrogen fertilizers and animal manure applied to soils, manure management in livestock farming is a key factor in ammonia emissions [3]. In particular, ammonia emissions originated in pig farms are an important factor to consider, as they represent a significant direct health factor in closed housing. It has been shown that high exposure to ammonia causes oxidative stress and injuries in lung tissue of piglets [4]. Since ammonia is also toxic to humans, it poses a potential risk for farm workers.

In order to mitigate the ammonia emissions, there are recommended thresholds for ammonia concentrations in pig barns reaching from 7 ppm up to 20 ppm for pigs and 7 ppm to 35 ppm for farm workers. Common NH_3_ concentrations in pig barns are highly dependent on the type of pig farming and feeding, the housing and ventilation system. Values ranging from 1.9 ppm up to 26.5 ppm have been reported. To meet the requirements, mitigation measures such as feeding strategies, e.g., reduced protein content, air filtration or manure treatment, e.g., solid-liquid separation, are available to reduce either ammonia emissions or exposure in pig barns [5]. However, more data with a high spatial and temporal resolution is needed for a detailed assessment of mitigation measures. Ammonia measurement instrumentation ideally requires high sensitivity to detect even small emissions. Furthermore, a high selectivity is required to withstand the many potential interferences from other gases present in agricultural scenarios such as a pig barn. Ideally these measurement methods should be easy-to-use, deliver real-time data of emissions and require low maintenance effort to be attractive for smart farming applications. Self-contained and low-cost sensors with a small form factor and the ability to be integrated into sensor networks are needed in multiple farms and areas of a barn in order to find ammonia sources and contribute to a better understanding of ammonia emissions [6,7].

Fourier transform infrared (FTIR) spectroscopy is a common method for ammonia emission monitoring especially in dairy barns. Typically, pumps and several hundred meters of tubes are installed in a barn to draw air from various locations, which is then analyzed over a period of several minutes. Despite the high sensitivity of this method, measurements are associated with great effort and costs for farmers [8,9]. Electrochemical (EC) sensors on the other hand are often used by farmers because there are inexpensive and easy-to-use handhelds on the market. Apart from the reduced sensitivity when compared to spectroscopy-based techniques, drift effects and a shorter lifetime of EC cells are reported [10]. Depending on the type of electrolyte used, the electrolyte is subject to be consumed during measurement, reducing its lifespan, or electrolyte poisoning may occur when exposed to the target gas [11]. However, the temporal and spatial resolution of ammonia emission monitoring using optical or EC-sensors does not serve to appropriately evaluate mitigation measures or to acquire ammonia data in agricultural scenarios.

Recently, metal oxide semiconductor (MOS) gas sensors have been widely studied, as they represent a useful alternative due to their high sensitivity, fast response time, high lifetime and low maintenance and costs [10,12]. MOS sensors are based on a heated semiconductor layer, which reacts by chemisorption with gaseous compounds in the ambient air, resulting in a change in electrical resistance. Whenever the temperature of the layer is changed, a new equilibrium between oxidation and reduction processes on the sensor surface is established over time. This phenomenon is used in temperature-cycled operation (TCO) where a temperature modulation of the sensor enables increased sensitivity to the gas compound to be measured. In literature, this is often referred to as a virtual multisensor or virtual sensor array [13,14]. A virtual sensor array is recently also used in the carbon nanotube approach by means of different metal contacts [15].

A comparison of MOS sensors to infrared absorption and electrochemical sensors in terms of different parameters for deployment in agricultural gas monitoring applications is shown in Table 1. While the highly sensitive MOS sensors are superior to the other two sensing principles in terms of response time, maintenance, cost and suitability for portable instruments, their low selectivity is a shortcoming.

Many studies on MOS sensors are conducted in laboratories to test and improve the sensitivity and selectivity to oxidizing and reducing gases. Schultealbert et al. show that the components of a gas mixture exhibit characteristic exponential relaxations in resistance depending on the TCO. They propose the chemisorption to be based on a single dominant ionosorbed species on the surface and a single step reaction with a reducing gas such as ammonia. The results show a very good quantitative determination of ammonia from 0.01 ppm up to 116 ppm based on the proposed estimation of time constants [16]. The controlled environment of the gas mixing systems is a laboratory environment, which is very valuable and essential for the feasibility of MOS sensors for ammonia monitoring. However, this environment does not correspond to a real-world scenario with cross sensitive gases.

Recent research demonstrates the potential benefits of using commercially available MOS sensors in agricultural scenarios. In [8] a sensor network is used in a dairy barn to measure gas concentrations of CO_2_, NH_3_ and CH_4_. The MOS sensors for ammonia monitoring are calibrated in a laboratory using ammonia in synthetic air. Data from ten sensors placed at various locations in the pig barn are compared with reference data from FTIR spectroscopy, showing a high correlation coefficient. The authors concluded that the MOS sensor is suitable for identification of relative changes in NH_3_ emissions. However, due to the naturally ventilated barn, ammonia exposure is significantly lower than typical levels in closed pig barns, with concentrations of 0.5–3.5 ppm. Although the MOS sensor data are acquired in real barn conditions, the cross sensitivities with respect to hydrogen sulfide or humidity, for example, are not addressed, as the sensors are calibrated under laboratory conditions.

The presence of cross-sensitive volatiles acting on the sensitive layer increases the complexity of chemisorption modeling to an extent that the use of artificial intelligence (AI) has to be addressed. The application of a virtual sensor array through the use of TCO enables the creation and optimization of fingerprints for a specific gas. By analyzing the relaxation processes at different temperatures, more information about the gas components is obtained. The temperature-dependent sensitivity of the MOS sensor for different gases, combined with corresponding reference values and the AI, enables the selectivity to be improved. In this work, a methodology for the design of a virtual MOS sensor array is proposed, that combines a solid state sensor model with a cyclic data-driven design approach. The solid-state sensor model provides the constraints in terms of temperature level, duration and number of temperature steps, while the data-driven design process provides the evaluation by training and validating machine learning algorithms in each cycle. The performance of the designed virtual sensor array is then evaluated in operational scenarios. Data from our previous work on ammonia monitoring (Section 2) is used to optimize the temperature modulation. This virtual sensor array is used to carry out measurements in mechanically ventilated pig barns for approximately two weeks as a proof-of-concept for the quantification of ammonia. The used sensors, the design approach of the virtual sensor array, as well as the experimental setup and the methodology for regression on ammonia concentrations are presented in Section 3. In Section 4, the results on the sensor array design and on the regression on ammonia concentration are presented. The impact of the sensor-to-sensor deviation and of a protective particulate matter (PM) filter membrane is assessed. Additionally, the performance of the regression models is analyzed in different pig barns to estimate the transferability to untrained environments. Section 5 discusses and contextualizes the results within a broader scope of ammonia monitoring in livestock buildings. Finally, the Conclusions (Section 6) summarize the key outcomes of this work, while Section 7 explores potential advancements for future research.

## 2. Previous Work

In previous work on ammonia monitoring in agriculture, our group analyzed classes with no, low, medium and high ammonia concentrations. For this purpose, four aqueous solutions with different ammonia concentrations were prepared by diluting a commercial 25% ammonia solution with distilled water using a dispenser. According to Henry’s law, the gas concentration in the headspace of the emitting surface of the aqueous solution is proportional to the ammonia concentration in the solution. The NH_3_ gas concentration range is estimated using a handheld electrochemical ammonia sensor (AR8500, Smart Sensor, Dongguan, China). The MOS sensor board is positioned in the headspace of a ceramic cup containing the aqueous solution. The setup has been placed in a fume cupboard, representing the micro-meteorological airflow conditions of real applications. Using the Bosch BME688 Development Kit with AI-Studio software (version 1.5.5) [17], different temperature cycles, the so-called heater profiles, are explored, training datasets are recorded, and algorithms are validated. A proof-of-concept of an innovative instrument to monitor and classify ammonia emissions is shown. The measurements show high accuracy in classification [18].

However, our previous work was conducted in a laboratory setting with classes of ammonia concentration without potential cross-sensitive gases that are available in pig barns like H_2_S or CH_4_. The BME688 sensor resistance outputs are influenced by these gases during ammonia emission monitoring in pig barns. The same is true for the parameter humidity, which is known to strongly influence the sensor resistance outputs. The objective of the current work is to use raw resistance values to train AI regression models for predicting ammonia concentration with optimized accuracy, while minimizing the influence of cross-sensitive gases and humidity.

## 3. Materials and Methods

### 3.1. Sensors

#### 3.1.1. Metal Oxide Semiconductor Gas Sensor

In this work, the Bosch BME688 development kit is used. The hardware consists of the sensor board with eight sensors, an Adafruit board with an ESP32 MCU, MicroSD card for data storage and a button battery for the real-time clock [19]. The low-cost platform allows developers to have direct access to the TCO and explore different temperature steps, representing different sensitivities of the gas component and creating a unique fingerprint. This offers the ability to design the virtual MOS sensor array and optimize the sensitivity and selectivity of the sensor for a specific application. This sensor has already been used by the scientific community in a variety of applications, including monitoring of air quality [20], bacterial growth [21] and food quality [22]. However, only very few works are dedicated to the opportunity to have free access to the TCO and optimizing temperature modulations. Often the standard configuration or even a constant operating temperature is used in previous works.

The BME688 sensor board shown in Figure 1 is equipped with eight 3.0 × 3.0 × 0.93 mm^3^ System-in-Package MEMS devices, which contain a MOS sensor together with additional environmental sensors for temperature, humidity and pressure. According to the manufacturer, the sensor-to-sensor deviation is ±15% for Indoor Air Quality index applications. The TCO is implemented by an application-specific integrated circuit (ASIC) executing temperature cycles, so called ‘Heater Profiles’ (HP), on the sensor. A Heater Profile consists of multiple temperature steps, at the end of which the resistance of the sensitive layer is measured. The sensor is operated by continuously repeating the HP. To reduce power consumption sleeping cycles without heating or resistance measurements can be added [23]. A trained model then runs directly on the sensor board in form of a self-contained system.

The BME688 development kit is powered via micro-USB cable. The data from the eight MOS sensors, the temperature, humidity and pressure sensor are stored on a microSD card during training data recording.

#### 3.1.2. Electrochemical Reference Sensor

In order to train regression algorithms, a reference sensor is needed, that meets the requirements for ammonia monitoring in terms of animal and farm worker health and welfare. In this work, the Dräger NH_3_ FL [24] electrochemical sensor is used. This EC sensor has four electrodes, including an auxiliary electrode in addition to the working, counter and reference electrodes used in most three-electrode EC sensors. The sensor has a range of operation of 1 ppm to 100 ppm with an accuracy of ±5% of the measured value or a minimum of ±1 ppm and is equipped with a dust filter. According to the manufacturer, the sensor withstands a permanent exposure of 50 ppm ammonia, which is essential for long-term measurements in a pig barn [24].

In addition to the ammonia sensor, the housing of the gas monitoring system X-node from Dräger also contains other sensors such as a temperature sensor, which is used in this study. The data from the sensors is sent to the cloud via the Long Range Wide Area Network (LoRaWAN), where it is available for pre-processing [25].

### 3.2. Experimental Setup

#### 3.2.1. Pig Barn Scenarios

This section describes the setup for regression learning on ammonia concentrations, carried out in a pig farm. Due to its animal welfare measures and publications, Prignitzer Landschwein GmbH (Brandenburg, Germany) is preferred for a proof of concept on ammonia emission monitoring in pig barns. The company has 1300 sows, 5500 rearing piglets, 7500 fattening pigs and 350 hectares of farmland. Automation techniques are used to improve welfare conditions in piglet rearing, such as the separation of manure and urine using a manure conveyor belt [26].

The data is recorded with the BME688 MOS sensor and the Dräger EC sensor as the reference. Artificial Intelligence (AI) models are trained to quantify the ammonia concentrations. In addition to NH_3_ and humidity, several other gases or volatile organic compounds (VOCs) are expected to be present in a pig farm and could affect the MOS sensor. Four locations with different type of pigs are selected for the measurements. All pig barns are in a closed building and the temperature-controlled ventilation system actuates fresh air inflow. The measurement is conducted during all day husbandry in pig barns, and the environmental conditions inside the barns (temperature, humidity, VOCs, etc.) are not influenced by our measurements. The training is deliberately carried out in four different environmental scenarios to increase the generalization capability for various environments. Table 2 provides an overview and Figure 2 gives an impression of the four pig barns.

#### 3.2.2. Measurement Setup

The measurement setup installed in each of the four barns is shown in Figure 3. A commercial, battery-powered EC reference sensor is positioned at the center of the suspended structure. Attached to this sensor is a 3D-printed platform designed to hold four development kits, each equipped with eight BME688 MOS sensors (Figure 1).

Due to limited space and ventilation in pig barns, particulate matter (PM) concentrations can reach high levels. The PM size distribution described in [27] informed the selection of a protective membrane. To shield the sensors from PM exposure, a Gore VE82029 filter membrane made of expanded polytetrafluoroethylene (ePTFE) is used. Sensors covered by this filter are identified by a black double-ring holder (see Figure 3).

The EC reference sensor transmits ammonia and temperature data to the cloud via LoRaWAN, allowing functionality checks throughout the 15-day measurement period. At the end of the study, all reference sensor data is downloaded as ground truth. The BME688 development kits are powered via micro-USB cables, and their training data is stored on microSD cards. Since real-time access to training data is not possible, all data is retrieved at the end of the measurement period for processing on an external computer.

Each barn’s measurement setup consists of one EC reference sensor and 32 MOS sensors, suspended approximately two meters above the pigs using cable ties. As per the EC sensor data sheet [24], no drift compensation was applied. This methodology generates a large dataset from four relevant pig barn environments, enhancing AI robustness for training, validation, and testing. Additionally, it enables analysis of sensor manufacturing tolerances and their impact on sensor-to-sensor variation. Further investigation is proposed to assess the influence of the protective membrane on sensor performance.

#### 3.2.3. Sensor Data Acquisition, Processing, Training and Deployment Phase

This paper details the training of the BME688 sensor in pig barn scenarios, Figure 4. It involves data pre-processing, temperature profile exploration, and AI model training and testing. The BME688 development kit, equipped with eight MOS sensors, is designed to collect extensive training data. This data is processed on an ESP32 microcontroller board and stored on a microSD card, as real-time sensor readings are not required during the measurement.

At the end of the 15-day campaign, raw MOS sensor data from the microSD cards are transferred to an external computer for pre-processing. Ammonia reference data, required for regression, is obtained from the EC sensor in the measurement setup. This reference sensor wirelessly transmits ammonia and temperature data to a LoRaWAN gateway, enabling real-time access via a mobile device. The ammonia data from the reference sensor serves as ground truth and is retrievable via the cloud for further processing. Both the MOS sensor data and the ground truth data are then used to train AI models with a large training data set in many relevant scenarios.

In the deployment phase, which is beyond the scope of this paper, the AI code is deployed onto a microcontroller within the deployment setup. This setup, intended for end-user measurements, consists of an application board and a shuttle board with one BME688 MOS sensor in case of strong footprint constraints, as shown in Figure 4.

### 3.3. Design of the Virtual MOS Sensor Array

This section outlines the methodology for designing the virtual MOS sensor array. It incorporates the standard solid-state model for metal-oxide semiconductor (MOS) sensitive layers and the time constants for ammonia detection, as reported in recent studies. This model is integrated with a data-driven approach to iteratively optimize the virtual sensor array through the training and validation of AI algorithms.

#### 3.3.1. Solid-State Sensor Model

When the sensitive layer of a MOS sensor is heated, oxygen from the atmosphere strongly oxidizes the sensor surface. Electrons become trapped at the grain boundaries of the polycrystalline semiconductor layer, leading to the ionosorption of O^−^ ions [28]. This process creates depletion zones and increases the energy barriers between grains, resulting in a higher electrical resistance of the layer. Tin dioxide (SnO_2_) is one of the most commonly used materials for the sensitive semiconductor layer, Figure 5a. The electrical resistance of the layer is determined by the energy barriers at the grain boundaries. In the presence of reducing gases, the surface of the semiconductor layer undergoes reduction as O^−^ ions react with the gases, releasing trapped electrons back into the grains. This reduces the depletion zones and lowers the electrical resistance of the layer, as shown in Figure 5b [13,29,30].

Semiconductor materials show an exponentially decreasing resistance with increasing temperature. Taking into account both the temperature dependency of semiconductors and the change in resistance due to chemisorption, the resistance of the SnO_2_ layer is expressed as:(1)$$R=R_{0}\;\cdot \;e^{\frac{E_{B}}{k_{B}T}}$$
where:$R_{0}$ is the basic resistance at room temperature,$E_{B}$ is the energy barrier affected by chemisorption processes,$k_{B}T$ is the thermal energy.

In this work, the MOS sensor operates using TCO, facilitated by the ASIC in the System-in-Package BME688. The temperature cycles consist of discrete steps, such as shown in Figure 6: T_HIGH_ and T_LOW_. Due to the small thermal mass of the MOS sensors micro-hotplate and sensitive layer, it achieves its operating temperature within 20–30 ms [23]. Consequently, heating, cooling, and the associated changes in semiconductor resistance, such as those depicted in Figure 6 (① to ②, or ③ to ④), are considered instantaneous. Reduction and oxidation processes at the sensor surface, caused by volatile gases, alter energy barriers and resistances due to chemisorption. Additionally, chemisorption is temperature-dependent, with higher temperatures leading to increased oxygen ionization. These processes are modeled using exponential relaxation dynamics, as shown in Figure 6 (② to ③, or ④ to ①). Each relaxation is characterized by a single time constant [29]. When the time constant is small relative to the duration of a temperature step, chemisorption reaches equilibrium, establishing a steady-state condition. However, for larger time constants, the resistance does not reach equilibrium during the temperature step. In applications with sub-minute sampling rates and trace gas concentration analysis, typical time constants often exceed the duration of a temperature step. As a result, the MOS sensor resistance in TCO operates predominantly in a non-equilibrium state [12].

The resistance after the decrease in temperature is described by:(2)$$R\left(t\right)=R_{min}+\left(R_{max}-R_{min}\right)\;\cdot \;e^{-\frac{t}{\tau }}$$

The time constant τ defines the relaxation from the starting resistance $R_{max}$ to $\frac{1}{e}\approx 36.8\%$ of $R_{max}$ related to $R_{min}$:(3)$$R\left(t=\tau \right)=R_{min}+\frac{1}{e}\;\cdot \;\left(R_{max}-R_{min}\right)$$

The presence of reducing or oxidizing gases in the atmosphere affects τ, which is utilized to detect or quantify the gas species. In [16], an estimate for quantifying τ is given after exposing a SnO_2_ MOS sensor to different NH_3_ concentrations with a temperature step from 450 °C to 150 °C. Given a relative humidity (RH) between 40% and 50% and different ammonia concentrations $c_{NH_{3}}$, τ can be estimated by:(4)$$\tau =\frac{1}{a\;\cdot \;c_{NH_{3}}}$$
where:a=0.03078 at 40% RHa=0.03224 at 50% RH.

In our previous work detailed in Section 2, various classes of aqueous ammonia solutions were prepared in the laboratory. Ammonia concentrations in the headspace above the solution were estimated using a highly noisy handheld EC ammonia sensor, ranging from approximately 1 ppm to over 100 ppm. These concentrations align with the specifications of our pig barns, which typically range from 1 ppm to 30 ppm. In the framework of the solid-state model presented above and using Equation (Equation 4), the expected relaxation time constants for ammonia in synthetic air are estimated, assuming ammonia concentrations (c_NH_3__) between 1 ppm and 100 ppm and a relative humidity (RH) of 50%. The resulting time constants are in a range of:$$\begin{array}{c} 0.31\;s\leq \tau \leq 31.02\;s \end{array}$$

In conclusion, the solid-state model provides valuable insights into the time constant (τ) of chemisorption relaxation processes for ammonia concentrations within the range relevant to pig barn applications. For ammonia concentrations of 30 ppm and 1 ppm, the steady-state resistance is achieved by 36.8% approximately after 1 s and 31 s, respectively. This suggests that the minimum duration for a single discrete temperature step should be in the order of seconds. Consequently, the total sample time, corresponding to the full temperature cycle of 10 steps, is about 10ths of seconds. The resistance of the sensitive layer is measured at different times in each temperature cycle, forming a sample with f features.

#### 3.3.2. Data-Driven Design

To enhance sensor performance, this work proposes combining the solid-state sensor model with a data-driven approach to design a virtual sensor array using discrete temperature steps. The methodology follows iterative cycles of design and redesign, with each cycle evaluated through training and validating AI algorithms. The process, based on the methodology outlined in [31], is illustrated in Figure 7.

Using the proposed model guidelines for step duration, sample time, and temperature levels, an initial sensor board configuration (Z_1_) is created. This configuration consists of *j* heater profiles (HP_j_) designed to collect sensor data under the target application scenario. Each HP comprises *f* steps, representing the *f* features of the gas-component fingerprint. The initial set is recommended to include a variety of step shapes, temperature levels, and durations. By training and evaluating AI models to predict gas concentrations for each heater profile, the best-performing heater profile (HP’_1_) is identified during the first design cycle. For classification tasks, accuracy—a measure of correct classifications divided by total classifications—is a commonly used performance metric. For regression tasks, the root mean squared error (RMSE) is proposed as a suitable metric. The characteristics of HP’_1_, such as its overall shape, number of temperature levels, step durations and temperature range are analyzed. Based on these findings and in alignment with the solid-state model guidelines and operational constraints, a new set of heater profiles (Z_2_) is developed for analysis in the next design cycle. This iterative process is repeated until the chosen performance metric demonstrates significant improvement in predicting the ground truth ammonia concentration.

### 3.4. Methodology for the Ammonia Regression in Pig Barns

This section describes the method for the ammonia measurements performed in four different pig barns used to train regression models. These measurements are used to investigate the performance of and the potential to transfer the designed virtual MOS sensor array to a realistic scenario. First, the treatment and processing of the data, as well as the regression models are introduced. Then, the division of the dataset into various subsets is described, which serves the investigation of multiple aspects of the applicability in real-world scenarios.

#### 3.4.1. Interpolation of Reference Data

The sampling rate of the MOS sensors for a single sample, i.e., f temperature steps, depends on the used temperature cycle. However, the EC reference sensors provide one NH_3_ value per minute. As the data of the reference and MOS sensors is not synchronous, the Piecewise Cubic Hermite Interpolation Polynomial function in Matlab is used for an interpolation of reference data. The value of the interpolated function is then taken at the corresponding mean time value of the f time stamps in one temperature cycle of the MOS sensor to get an estimated reference value for this exact sample.

#### 3.4.2. Regression Models

After the measurement, the gas sensor samples are transferred to an external computer, where regression algorithms are trained. Two regression models are considered: a Bagged Trees algorithm with 30 learners and a compact Neural Network consisting of three layers, each with ten neurons. Especially the Neural Network is suitable for self-contained systems, where the code is ported to the microprocessor of an embedded system. Model training and testing are conducted using the MATLAB Regression Learner application (version 24.2) [32]. Only pre-configured models are used, so there is no hyperparameter tuning with additional validation datasets.

#### 3.4.3. Metrics for Evaluation

The trained and tested regression models are evaluated based on their performance, using root mean squared error (RMSE) as the primary performance metric. To facilitate comparison with the actual ammonia concentrations in the pig barns, the normalized RMSE (NRMSE) is computed by dividing the RMSE by the mean reference concentration of the corresponding barn(s) over the whole measurement session. Residual plots are employed to analyze prediction errors across the concentration spectrum. Additionally, to assess the potential for qualitative ammonia trend monitoring—such as detecting daily patterns—response plots are utilized, illustrating the true and predicted values over time.

#### 3.4.4. Data Subset Division

The fifteen-day-dataset collected in the pig barns is divided into multiple subsets. Using different combinations of subsets for training and testing of regression models, multiple aspects of the applicability, regarding the robustness of the algorithms of the MOS sensor based ammonia monitoring in pig barns are assessed. The data division is illustrated in Figure 8.

The results and discussion on the ammonia regression performance address three key challenges in our experimental setup: sensor-to-sensor deviation, the impact of the filter membrane, and the environmental factors represented by four pig barns with varying ammonia levels and cross-sensitive gas emissions. To ensure a focused analysis and minimize overlapping effects, each aspect is evaluated sequentially in the following order:Influence of the Sensor-To-Sensor Deviation: First, the data from one measurement location that contributes a large amount of data and covers a wide ammonia concentration range is selected. This dataset contains data from *n* sensors S_n_. The dataset is divided into D_nF_ and D_F_, each containing data from sensors without or with filter membranes, respectively, Figure 8. D_nF_ is split into the data originated from the *i* individual sensors and used to assess the influence of the sensor-to-sensor deviation on the ammonia regression performance. To achieve this, a trained model is first tested with a regular test dataset and then tested again with an additional test dataset T_SD_, coming from sensors which were not included in the training data. Considering the relatively small number of sensors, a *k*-fold cross validation is proposed to use here and is illustrated in Figure 9.The variable *k* defines the number of folds, in which D_nF_ is divided and is selected to have two sensors in each T_SD_ test set. For every fold, the RMSE of the two regression models is first calculated for the 25% holdout test data of i−2 sensors. Second, the RMSE for the additional test dataset T_SD_ from sensors excluded from the training dataset is determined. The mean values and the standard deviations (Std. Dev.) of the RMSE values across all folds and for every model and test method are calculated. Using this methodology, the influence of the sensor-to-sensor deviation and the transferability of the models to other sensors of the same type are assessed.Impact of the Filter Membrane: To investigate the influence of the filter membrane, the models are separately trained with D_nF_ or D_F_, respectively. For the test, 25% of D_nF_ or D_F_, respectively, is used. The resulting RMSE values for the models trained and tested with data originating from sensors with or without filter membranes are compared, without consideration of individual sensor datasets.Transferability to Other Environments: Finally, the influence of different training environments, i.e., pig barns, is assessed. The whole dataset is split according to the locations A, B, C and D without consideration of the individual sensors or filter membranes. Then, four data splits with training and test datasets, wherein three locations are included in each training dataset D_wxy_ and the fourth location is exclusively present in the test dataset T_z_ are created. Every D_wxy_ is again divided into 75% training data and 25% test dataset. Although the effects of the sensor-to-sensor deviation and the filter membranes overlap with the influence of the environments at this point, the variability of the results can be used to provide a qualitative assessment of the influence a different training scenario has on the regression. This offers an insight into the capacity of the models to be transferred to a pig barn not included in the training process, if the training was performed in various environments.

## 4. Results

### 4.1. Results on the Design of the Virtual Sensor Array

The proposed Virtual Sensor Array Design provides insights from solid-state physics combined with AI-driven evaluation and optimization of temperature cycles applied to the sensor. These temperature cycles, referred to as Heater Profiles (HP) in this work, are characterized by the number of steps, temperature range, temperature levels, and the duration of each step. The number of steps corresponds to the number of features f used for training the AI-algorithm. For the sensor employed in this study, the manufacturer specifies ten steps. Similarly, the temperature range is constrained by the sensor’s material properties, manufacturing processes, and the limitations of its analog-to-digital converter. The BME688 sensor operates within a temperature range of 50 °C to 400 °C.

The choice of temperature levels and the duration of each step is guided by the relaxation time constant of temperature-dependent chemisorption processes at the sensitive semiconductor surface, as described in the solid-state model section (see Section 3.3.1). For each gas mixture, the time required for the sensor’s resistance to reach steady-state conditions depends on the gas composition and concentrations. In pig barns, ammonia concentrations typically range between 1 ppm and 30 ppm, with significant fluctuations in humidity due to temperature-controlled airflow and the presence of large amounts of VOCs from pig metabolism. The estimated relaxation time constants for an ammonia concentration of 1 ppm and 30 ppm in synthetic air with 50% relative humidity are 31 s and 1 s, respectively [16]. Consequently, step durations in the order of a few seconds are sufficient for prediction of ammonia. This results in a high sampling rate for the ammonia monitoring with less than one minute per sample.

The development kit board, equipped with eight BME688 sensors, enables the simultaneous exploration of up to four HPs, significantly enhancing statistical analysis and reducing development time. This capability is used, allowing the initial board configuration Z_1_, as shown in Figure 7, to examine a set of four HPs. Details of this configuration, Z_1_, are illustrated in Figure 10.

HP354: With a sampling time of 10.78 s, this profile features three distinct temperature levels, transitioning sharply from a high temperature of 320 °C to a low temperature of 100 °C.HP301: Sharing a similar structure with three levels but utilizing longer sampling times (18.34 s).HP411: With a sampling time of 24.64 s, this profile incorporates high-temperature spikes, allowing the evaluation of their effects on sensor performance.HP501: This profile spans five temperature levels over a sampling time of 26.88 s, characterized by smaller temperature drops between levels.

These heating profiles provide a diverse range of sensitivities to ammonia and distinctive fingerprints, enabling a comprehensive evaluation of sensor performance.

According to the design flow outlined in Figure 7, the next step is to train and evaluate the AI algorithm. This process builds on the ammonia classification tasks detailed in the section on our previous work (Section 2). Sensor data was recorded in the laboratory using aqueous ammonia solutions. Ten temperature steps serve as features for training a neural network to classify emissions from four distinct ammonia solutions. Ground truth data was provided by a handheld EC ammonia sensor AR8500 from Smart Sensors. The BME688 development kit includes eight sensors configured with the initial sensor board setup (Z_1_) featuring four HPs as described earlier. Figure 11 visualizes the resistivity data for a single sensor from two HPs (HP354 and HP501). The black dots represent the mean resistance averaged over time during exposure to the high ammonia class. The dashed line depicts the resistance behaviour based on the solid-state model. For HP354, the average step duration is 1.2 s, compared to 2.7 s for HP501.

Given exponential relaxation time constants in the order of seconds for this ammonia exposure, steady-state conditions are expected during the measurements, as shown in Figure 11. Furthermore, the resistance profiles reveal whether reduction or oxidation processes dominate following instantaneous temperature steps. For example, in HP354, the resistance decreases over time at 100 °C and 200 °C, indicating predominant reduction processes. Conversely, the resistance increases at 320 °C suggests that oxygen chemisorption outweighs reduction by ammonia.

For each HP, a neural network was trained for classification of the solutions. Given the balanced nature of the dataset, accuracy is proposed as the most suitable performance metric for evaluation. The results of the initial design cycle are presented in Table 3, where HP301 achieves the highest accuracy. However, with the exception of HP501, all measurements are affected by a clipping effect. This phenomenon arises when resistance values, at extremely low or high temperatures, fall outside the full-scale output of the sensor due to the limitations of the internal analog-to-digital converter.

Considering the close proximity of accuracy values in Table 3 and the reduction of data volume due to clipping, HP501 is selected as the best-performing HP’_1_ of the first design cycle.

According to Figure 7, the second set of HPs, referred to as Z_2_, is designed to align with the structure of HP501. In addition to HP501, HP502 is included in the design cycle due to its identical temperature steps but extended step durations, resulting in a sampling time approximately 10 s longer than HP501. The extended step durations of HP502 are particularly beneficial for predictions in the low concentration range, where higher time constants are expected. Both HP501 and HP502 share a temperature range of 100 °C to 320 °C.

Additionally, HP503 and HP504, which feature the same step durations but expand the temperature range to 70 °C–350 °C, are included in the evaluation. However, data from HP503 and HP504 exhibit clipping effects at the 70 °C steps and are therefore excluded, as previously explained. The evaluation results of Z_2_ are summarized in Table 3. Among the profiles, HP502 demonstrates the highest accuracy and is ultimately selected as HP’_2_ and the temperature cycle to be used in the regression measurements in the pig barns.

### 4.2. Results on Ammonia Monitoring in Pig Barns

#### 4.2.1. Pig Barns Scenarios

In this section, an overview of the environmental specifications and the data provided by the MOS sensors and the EC reference sensors in the four pig barns is given. The ammonia data recorded with the EC sensor and the temperature over the whole measurement session (from 16 July 2024, 05:14 pm to 31 July 2024, 03:27 pm) are illustrated in Figure 12. Overall, the range of NH_3_ concentrations recorded in the four pig barns varies significantly. In location B, C and D, the ammonia concentration is mostly below 10 ppm, while higher NH_3_ levels are observed in location A. Daily fluctuations in NH_3_ concentrations, which correlate with the temperature, can be seen, particularly in location C. The ammonia concentration peaks at low temperatures (during the night), as the temperature-controlled ventilation system provides reduced air exchange at these times.

During the measurement process, some of the BME688 sensors have failed to function, presumably as a consequence of contamination by dirt, dust, flyspecks and other contaminants. The extent of this effect differs between the pig barns, resulting in unbalanced datasets for the four measurement locations. A visual inspection of the sensor setups after the measurement session confirms the assumption that the presence of contaminants varies considerably between the pig barns. In Figure 13a, two dirty sensor setups from different locations after the recording are shown. The percentage of sensor data from each location in relation to all data is depicted in Figure 13b. A total of 2,271,587 samples of all MOS sensors combined are recorded.

#### 4.2.2. Impact of the Sensor-to-Sensor Deviation

Since location C provides a substantial number of samples and covers a wide concentration range, this pig barn is selected for the assessment of the sensor-to-sensor deviation. From the 16 sensors without a filter membrane in location C, two sensors failed. The data of the 14 remaining sensors form the dataset D_nF_. The cross validation method is selected with k=7 as the number of folds. In every fold, 12 sensors are used for training and testing and two sensors are excluded from training and used as additional test data (T_SD_) to provide a measure for the sensor-to-sensor deviation.

The values for the mean (normalized) RMSE and the standard deviation for the regular holdout test data and the test with data from sensors excluded from training T_SD_ are shown in Table 4. With test data from all sensors, the Bagged Trees model performs better than the neural network. The standard deviation of the RMSE values for both models is relatively low. When tested with full non-trained sensor data T_SD_, the mean RMSE increases for both models. The neural network model achieves a lower RMSE compared to the Bagged Trees model and the standard deviation is much higher in comparison to the regular holdout test data.

#### 4.2.3. Impact of the Filter Membrane

The impact of a PM filter membrane on the ammonia estimation is evaluated by training two models with datasets from sensors in location C with filter and without filter, respectively. The results are shown in Table 4. The RMSE values of the models trained with data without filter membranes D_nF_ are comparable to the results of the cross validation with holdout test data from the sensor-to-sensor deviation results (Section 4.2.2). When trained and tested with sensors equipped with a filter membrane (D_F_), both the neural network and the Bagged Trees models show lower RMSE values. Using the filter membrane, the NRMSE of the models is below 20%.

In Figure 14, the residual plots of the neural networks trained on each dataset are presented. In general, the absolute errors increase with increasing predicted NH_3_ values. The residuals are symmetrically distributed around zero.

Figure 15a displays the predicted response over the true (interpolated) concentrations of the neural network (trained with D_F_) with test data of one MOS sensor over a period of 13 days to visualize the progression over time in the pig barn C. Overall, the predicted response captures the daily pattern in ammonia concentrations influenced by the temperature-controlled airflow-intake. The errors shown in Figure 15b increase especially when the ammonia concentration reaches high values during the night.

#### 4.2.4. Environmental Impact

Table 5 shows the results of the investigation of the influence of different training scenarios. For each data split, three barns are included in the training dataset D_wxy_ and one represents the additional test dataset T_z_. The mean ammonia concentration in the datasets vary significantly.

When tested with holdout test data from the training dataset, the RMSE values are in the order of magnitude of the results in the previous sections. When being tested with data from different pig barns, the RMSE values increase. The intensity of this effect varies depending on the different pig barns and is most emphasized for the models trained with D_BCD_ and tested with data of location A (T_A_). However, the NRMSE is the lowest for this data split, because of the high NH_3_ concentration in Location A.

In Figure 16, the residual boxplots of two models (D_BCD_/T_A_ and D_ACD_/T_B_) are shown as an example. Both plots show a strong bias and heteroscedasticity. The distribution of the interpolated ammonia concentration values varies strongly between the two data sets.

## 5. Discussion

Following a discussion of the Virtual MOS Sensor Array Design, the measurements conducted in the pig barns are examined below. This includes a more detailed overview of the specific conditions within the pig barn environments. Key instrumentation parameters, such as sensor-to-sensor deviation and sensor packaging with PM filter membranes, are analyzed, along with the measurement scenarios across four different pig barns within the same farm.

### 5.1. Virtual MOS Sensor Array Design

In this study, the virtual MOS sensor array is optimized for classifying ammonia emissions from aqueous solutions in a laboratory setting using a neural network. The concentration range aligns with typical ammonia levels of up to 20 ppm in pig barns. Two design cycles are conducted to demonstrate the methodology, which integrates a solid-state sensor model with a data-driven design algorithm.

In the first design cycle temperature ranges, heating durations and temperature modulation shapes based on the underlying solid-state sensor model is explored. Clipping effects due to the operational range of the measurement device are taken into account, especially at lower temperatures, where the resistance of the sensitive layer is very high. Based on the results of this cycle, a second design cycle is conducted, integrating four HPs, matching the shape of the best-performing HP of the first cycle. The best HP of this second cycle features longer heating durations, which is crucial for the detection of lower NH_3_ concentrations, according to the given MOS sensor model. For a next third design cycle, it is proposed to increase the step durations even further, taking into account the sampling rate required for the application. The integration of additional measurement points in between the temperature steps is suggested to obtain more information about relaxation after a change in temperature.

The resulting ammonia monitoring sampling rate is determined by the duration of a full temperature cycle, which lasts a few tens of seconds. This confirms the feasibility of achieving a sampling rate of more than one sample per minute using the MOS-sensor BME688. The proposed virtual array sensor design is trained with AI-regression models and tested in representative pig barns.

### 5.2. Pig Barns

The ammonia measurements in this study are conducted over two weeks in four different pig barns, housing pregnant sows, 6- and 8-week-old piglets, and 270-day-old sows. These barns are equipped with temperature-controlled airflow systems to enhance animal welfare. During the summer, the poorly insulated barns tend to overheat. To mitigate heat stress, ventilation systems are activated, particularly during midday, improving air quality and reducing ammonia levels. In barns housing pregnant sows, the ammonia concentration is significantly higher than those recorded in the other barns. Since ammonia forms when urine and manure mix, differences in the pig excretions, along with environmental factors such as ventilation, are potential reasons for this. Strong daily fluctuations are observed characterized by ammonia peaks at night, due to the non-activated airflow system at lower temperature levels. The ammonia concentration ranges and the daily variations measured in this study align with typical pig barn climate conditions [5,34].

In addition to ammonia, various VOCs and odors are present in all four barns. We have smelled these differences firsthand during sensor installation in the barns. A more detailed discussion on cross-sensitive and interfering gases is provided below.

### 5.3. Impact of Sensor-to-Sensor Deviation

The manufacturing tolerances of MOS sensors, which consist of a membrane, micro-hotplate, and sensitive layer, impact raw sensor data and consequently the training of models [35,36]. To evaluate the effect of sensor-to-sensor deviation, data from sensors is excluded during model training. A study conducted in a 270-day-old sow barn, where the mean ammonia concentration is 6.48 ppm, reported mean NRMSE values of approximately 30% when using test data from the excluded sensors, compared to around 20% when using test data from sensors included in the training. The observed 10% increase in NRMSE when models are being tested with data from sensors excluded from the training is surprisingly small. Overall, the sensor-to-sensor-deviation is within an acceptable range for deploying the BME688 sensor in a pig barn environment.

Methods for compensating the effect of the manufacturing tolerances on the sensor-to-senor deviation of MOS sensors using calibration models, which are also trained with a high number of sensors and can be applied to a new sensor, are described in [36].

### 5.4. Impact of the Filter Membrane

When MOS sensors are equipped with a filter membrane, the trained models demonstrate improved performance compared to training with data from MOS sensors without a filter. This improvement is attributed to the protection of the BME688 against soiling, particularly from flyspecks, which affect the sensitive layer of the sensor in the absence of a filter membrane. During sensor installation, we observed a high number of flies, especially in the housing for the 6-week-old piglets (location B). At the end of the experiment, a substantial data loss is evident, most notably in the 6-week-old piglet housing. The accumulation of flyspeck contamination on the unprotected areas of the sensor board likely contributed to this data loss.

While the residuals of the trained models are symmetrically distributed around zero, absolute errors increase with rising predicted ammonia levels, particularly above 10 ppm. A visualization of the predicted and actual values over a 13-day measurement period shows that the predicted response closely follows the daily ammonia concentration patterns influenced by the temperature-controlled airflow intake. However, errors become more pronounced when ammonia concentrations reach extreme levels at night due to the lower number of training samples available at these higher concentrations.

### 5.5. Environmental Impact

When trained models are tested on data from barn locations which are not included in the training set, RMSE values vary significantly between barns. Residual plots reveal a strong bias, indicating that the models are not sufficiently adapted to the tested environments. This study, along with human sensory perception, confirms that even within the same pig farm, different barns present distinct training scenarios due to variations in cross-sensitive gases. These differences likely become even more pronounced when comparing barns across different pig farms, as factors such as feeding strategies and husbandry conditions further influence gas compositions. In addition to variations in ammonia concentration and humidity, the total VOC concentration and composition differ between environments [37].

### 5.6. Implications

While this work provides insight into several pig barn specific aspects of MOS sensor based ammonia monitoring, there are several implications related to humidity and cross sensitive VOC gases and PM filter effects.

In order to increase the variety of measurements and thus the robustness of the AI, four different pig barn scenarios are used for AI model training over a period of 15 days. In the current work, raw resistance data from the MOS sensor are used as input to train the models. However, humidity and cross-sensitive VOC gases are known to significantly affect the sensitivity and selectivity of the MOS sensor. Humidity sensor data is available from the System in Package BME sensor. In future work, it is recommended that the data from the BME688 integrated temperature, humidity and pressure sensors be used in advanced calibration models such as [38,39]. The aim is to reduce the potential influence of humidity and cross sensitive gases that are not eliminated by the intensive AI model training in different environments.

In the current work we recommend the use of a PM filter. However, the influence of the filter on ammonia sensitivity and latency issues is unclear. Although the use of a PM filter membrane improves quantification performance, it represents a diffusion resistance that may cause a response delay in the detection of changes in ammonia exposure. However, although this is likely to be a minor drawback, the effect of the filter as a diffusion barrier and retarder has to be investigated in future work.

## 6. Conclusions

This study presents a novel approach to design a virtual MOS sensor array for ammonia monitoring in pig barns, demonstrating its feasibility in real-world conditions. The method combines solid-state sensor modeling with a data-driven design, iteratively training predictive models to optimize performance. The optimized system achieves a sampling rate of over one sample per minute for ammonia concentrations between 1 and 30 ppm, ensuring suitability for continuous monitoring with a single BME688 sensor.

Field measurements across four pig barns over two weeks reveal significant ammonia variations influenced by barn type, ventilation, and pig metabolism. Pregnant sow barns show the highest levels, likely due to increased urine and manure production, with concentrations peaking at night when airflow systems are inactive. The presence of various VOCs further highlights the complexity of livestock environments. Sensor-to-sensor deviation is assessed by excluding individual sensors during model training, showing a 10% increase in normalized RMSE. This indicates the BME688 sensor system to be robust despite sensor manufacturing tolerances. A filter membrane improves model performance by reducing contamination of the sensitive layer. However, contamination of the unprotected parts of the sensor boards, particularly from flyspecks, causes data loss. This emphasizes the need for filters to enhance accuracy of models and a protection of the electronics for data availability. Testing trained models on barns excluded from the training set shows significant RMSE variations, highlighting the necessity to improve the training data set by training in various pig barn scenarios with different background situations, like feeding strategies and husbandry practices.

Overall, the BME688 MOS sensor effectively quantifies NH_3_ in high interference environments with sufficient accuracy for pig barn monitoring. Its low cost allows large-scale deployment across multiple farms and high spatial resolution of ammonia monitoring in barns, supporting the identification of emission sources. The high sampling rate enables tracking of ammonia fluctuations, validating mitigation strategies such as manure-urine separation and improved air quality measures for animals and humans.

## 7. Future Aspects

Future work will focus on further optimizing the design of virtual MOS sensor arrays. This includes research in temperature-cycled operation and iterative design improvements. Feature analysis tools will be utilized to fine-tune individual temperature steps and step durations. Additional redesign cycles will be implemented to enhance performance.

The results indicate that environmental differences in various pig houses significantly impact model training, emphasizing the need to expose sensors to diverse scenarios to ensure consistent performance across different conditions. The low cost of the sensors enables widespread deployment across multiple barns. An increased and diverse training dataset will enhance the generalization of regression models, making them more robust for inference in unseen environments.

In addition, it is important to investigate the long-term stability aspects of the BME688 in high humidity and high ammonia environments. Our work focuses on ammonia monitoring in pig barns, where the sensors are exposed to a maximum of 30 ppm ammonia. In general, MOS sensors are known to have several years of surface and bulk properties stability in ambient air. However, the main reason for long term instability is the change in metal oxide parameters caused by microstructural and morphological changes in the sensing layer when exposed to high concentrations of gases and humidity. As the BME688 is a sensor system in a package solution with on-board gas, humidity, pressure and temperature sensors, model-based approaches to compensate for potential ageing and drift effects will be investigated in future. Furthermore, the effect of the filter especially in terms of response time lag due to its diffusion resistance is to be analyzed in future.

Another key area for future research is exploring different machine learning algorithms to improve ammonia quantification. Incorporating a broader range of models and optimizing hyperparameters will be crucial in enhancing model generalization. Beyond the pig barn scenarios, the proposed virtual sensor array design approach is well-suited for agricultural applications such as fertilizer monitoring. It can be employed to compare and optimize different fertilizer treatments and application methods, contributing to strategies for reducing fertilizer use without compromising crop yield.

## Figures and Tables

**Figure 1 sensors-25-02617-f001:**
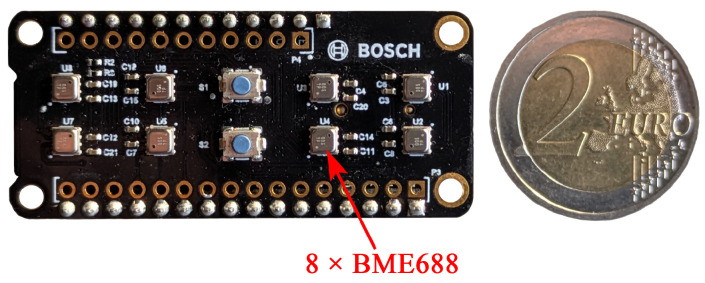
BME688 development kit with eight BME sensors.

**Figure 2 sensors-25-02617-f002:**
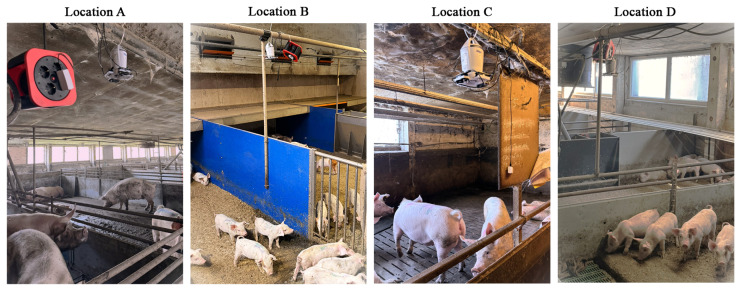
The four pig barns (A, B, C, D) with the measurement setups.

**Figure 3 sensors-25-02617-f003:**
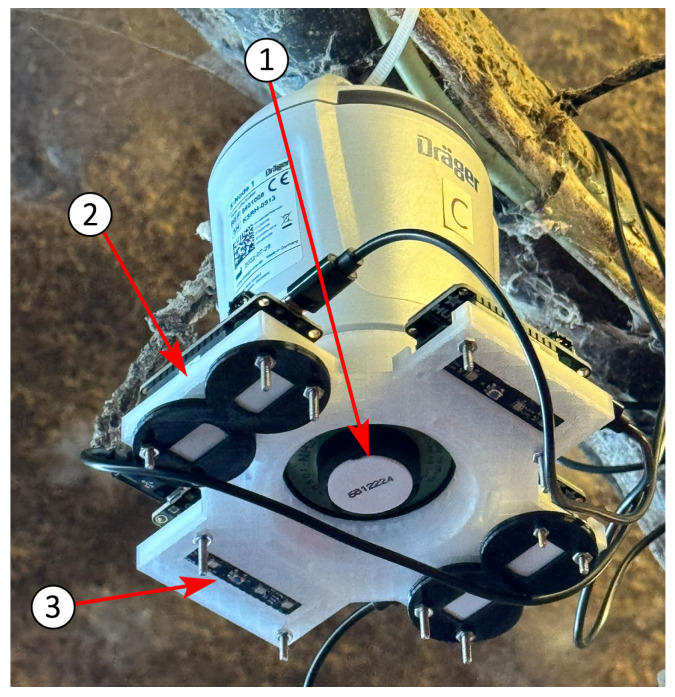
Setup of the sensors for training in the pig farm. EC reference sensor ①, BME688 sensor board with PM filter membrane ② and without PM filter membrane ③. The BME688 development kits are powered with micro-USB cables.

**Figure 4 sensors-25-02617-f004:**
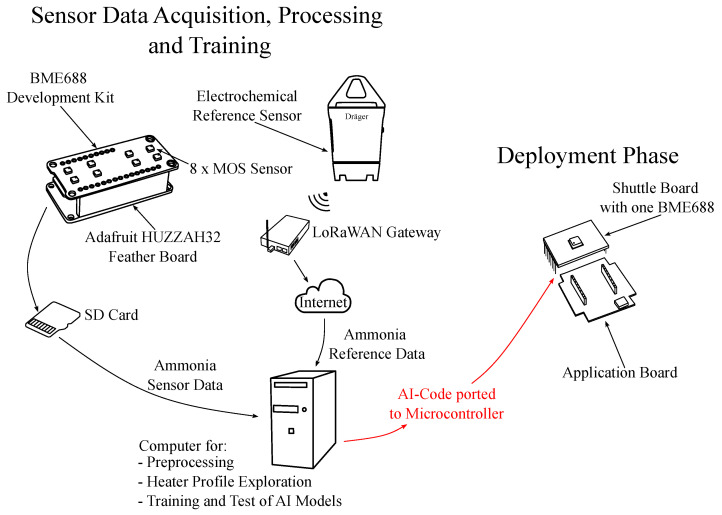
Sensor data acquisition and processing. On the left side, the sensor data acquisition for the training is illustrated, as conducted in this work. The deployment phase for future BME688 application is shown on the right side.

**Figure 5 sensors-25-02617-f005:**
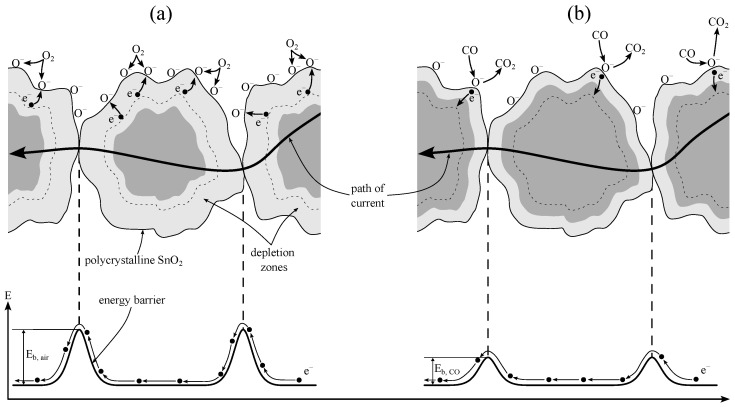
Solid-state model at grain-grain boundaries of the sensitive semiconductor layer. (**a**) The ionosorption of O^−^ at the surface leads to increased depletion zones and thus to higher energy barriers E_b, air_ at the grain-grain boundaries. (**b**) Reducing gases, e.g., CO, react with the ionosorbed oxygen to CO_2_, causing electrons to flow back into the material, leading to lower energy barriers E_b, CO_. Adapted from [13].

**Figure 6 sensors-25-02617-f006:**
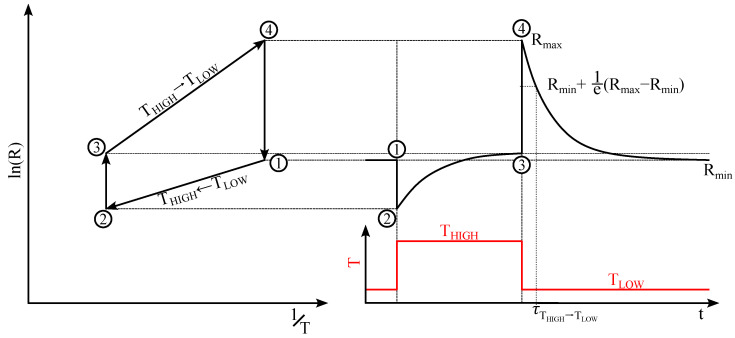
Schematic visualization of the states and the resistance of a MOS sensor shown as an Arrhenius plot. From ① to ② the sensitive layer is heated instantaneously, which results in a decrease of the resistance. From ② to ③, chemisorption takes place and the layer is driven towards a new state of equilibrium in ③. Then, from ③ to ④, the operating temperature is decreased, resulting in an instantaneous increase of *R*, followed by a process at the sensor surface to decrease the amount of oxygen ionized, which is described by the time constant τ. $R_{min}$ describes the steady state resistance at the equilibrium at $T_{LOW}$. Adapted from [29].

**Figure 7 sensors-25-02617-f007:**
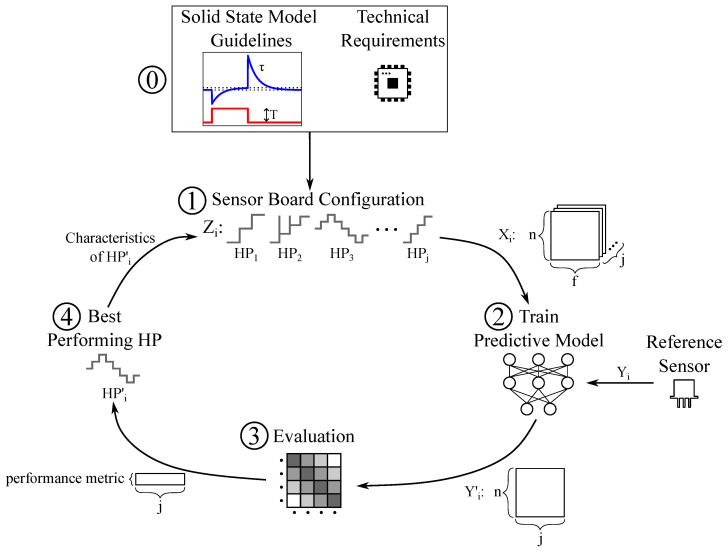
Design of the virtual MOS sensor array. ⓪ Definition of a suitable solid-state model and specifications for the MOS sensor. ① Definition of a set $Z_{i}$ of *j* heater profiles (HP) to be explored with conclusions from ⓪. With this configuration, sensor data $X_{i}$ is collected in relevant scenario. $X_{i}$ consists of a n × f matrix for every HP. The variable ‘n’ represents the number of observations, while ‘f’ represents the number of features per HP. ② Training of *j* supervised machine learning models to predict gas species with ground truth information $Y_{i}$ provided by a reference method. The models output a n × j matrix $Y_{i}^{\prime }$ representing gas species predictions for every heater profile. ③ Selection of a suitable performance metric depending on the used model and evaluation of the trained algorithms. This provides a performance metric for every *j*. ④ Selection of the best-performing heater profile HP’_i_ according to the defined performance metric. The properties of HP’_i_ are analyzed with information from the solid-state model guidelines in terms of duration of temperature steps and levels to create a new set of heater profiles. Return to step ①.

**Figure 8 sensors-25-02617-f008:**
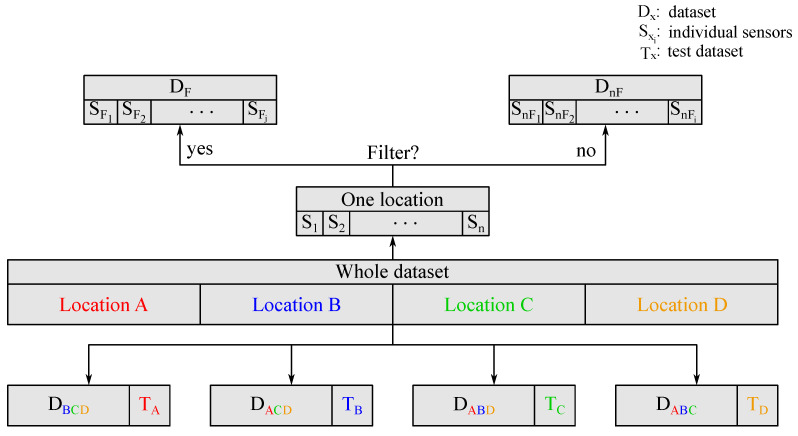
This figure shows how the recorded data from the pig barns is divided into multiple datasets D_x_. The indices mark the subgroups, where ‘F’ and ‘nF’ represent ‘Filter’ and ‘no Filter’. In the lower part of this figure, the indices specify the pig barns. A color is assigned to each barn for allocation.

**Figure 9 sensors-25-02617-f009:**
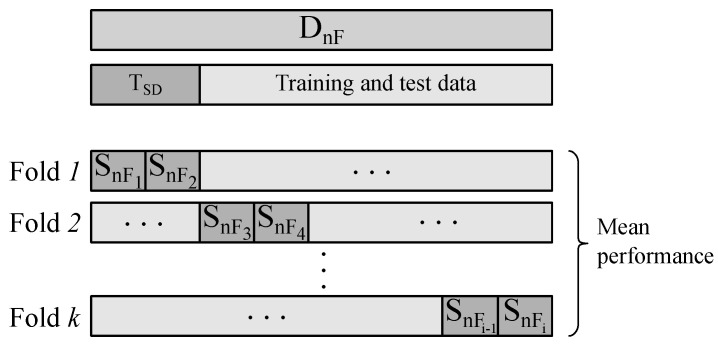
Cross validation used to investigate the sensor-to-sensor deviation. The data is split up into *k* folds, where in every fold, the data of two individual sensors is present in the sensor-deviation test dataset (T_SD_). The remaining data is further randomly split into 75% training data and 25% regular test data.

**Figure 10 sensors-25-02617-f010:**
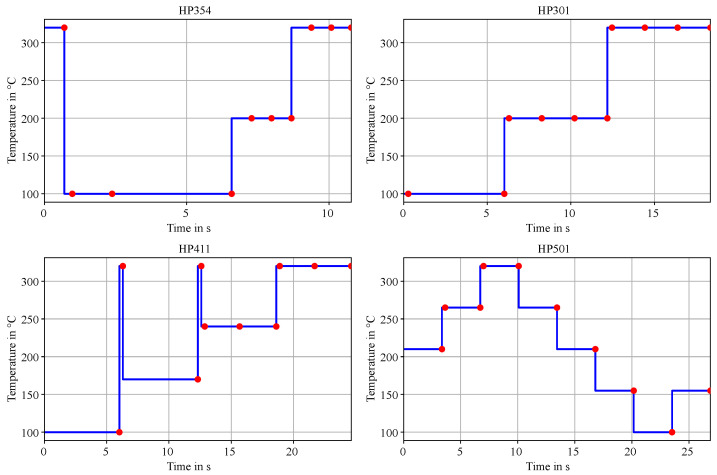
The initial set Z_1_ for the first design cycles containing four Heater Profiles (HP) given by the Bosch software BME AI-Studio [17,33]. The blue line represents the target temperature of the micro hotplate, and the red dots are the measurement points.

**Figure 11 sensors-25-02617-f011:**
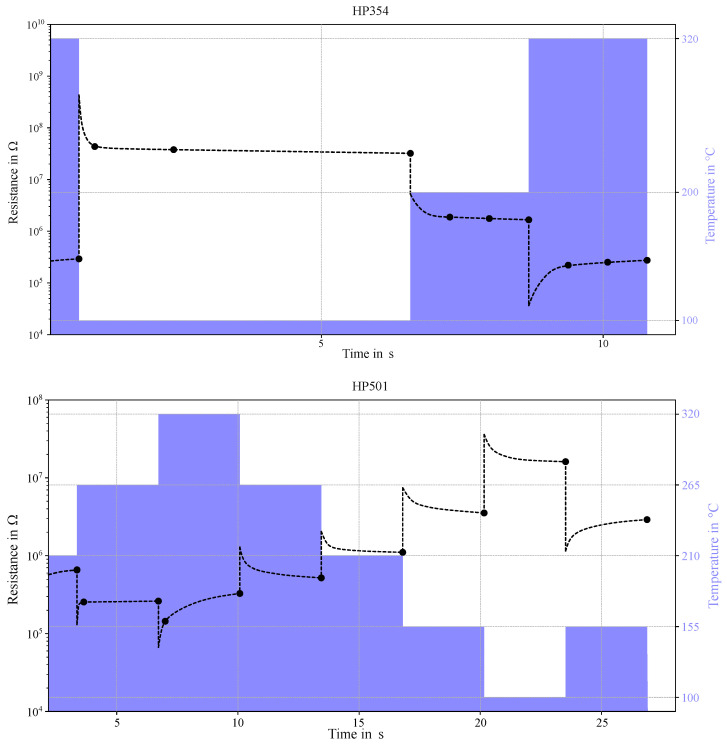
This figure illustrates the temperature and resistance behavior of two HPs with differing sample times. The black dots represent the mean resistance values averaged over the high ammonia exposure period. The dashed line depicts the resistance behavior predicted by the solid-state model, while the purple shaded areas indicate the cyclic temperature operation of the MOS sensor.

**Figure 12 sensors-25-02617-f012:**
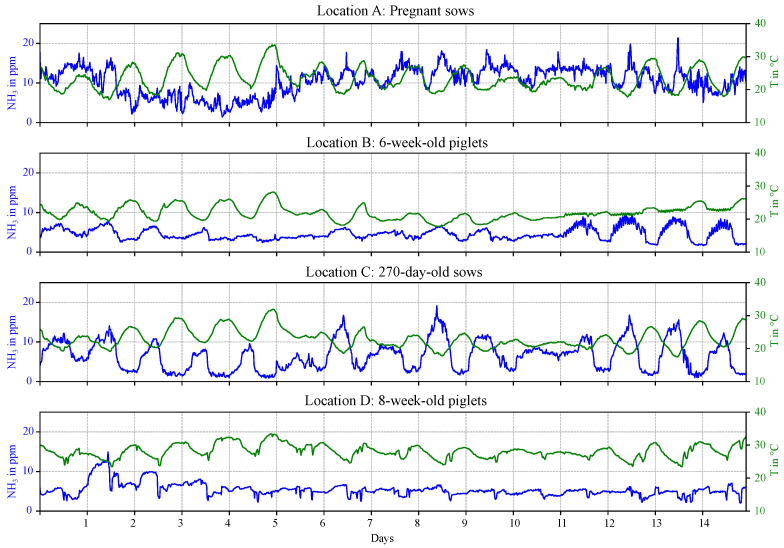
Ammonia concentration recorded by the EC reference sensor (blue line) and temperature in the pig barns (green line) during the fifteen days of measurement.

**Figure 13 sensors-25-02617-f013:**
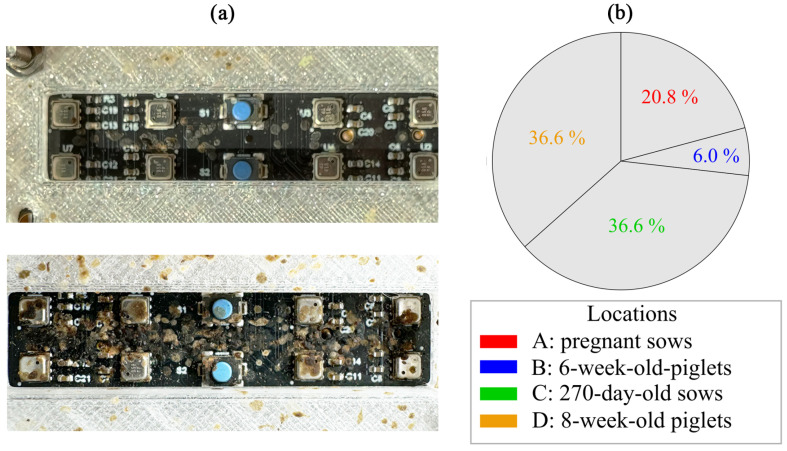
This figure shows a comparison of two dirty measurement setups (**a**) and the contributions of the MOS sensors of the four measurement locations to the whole dataset (**b**).

**Figure 14 sensors-25-02617-f014:**
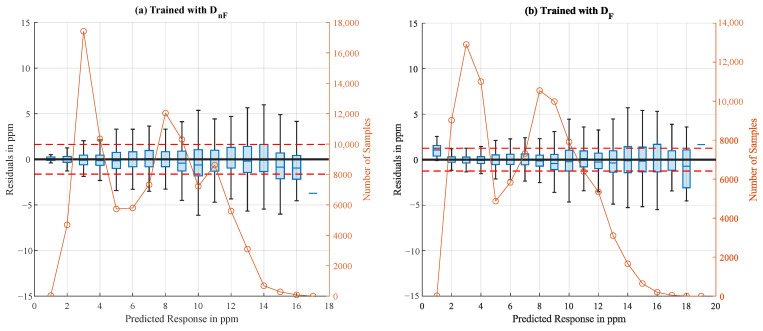
Boxplots of residuals from the neural networks trained with D_nF_ (**a**) and D_F_ (**b**), respectively, each tested with 25% holdout test data. The predicted values (abscissa) are grouped in bins with a width of 1 ppm, each represented as one box. The first box contains predicted values from 0 to 1 ppm, the second from 1 to 2 ppm, and so on. The orange circles represent the number of test samples (interpolated data) for each box, while the dotted red line shows the RMSE.

**Figure 15 sensors-25-02617-f015:**
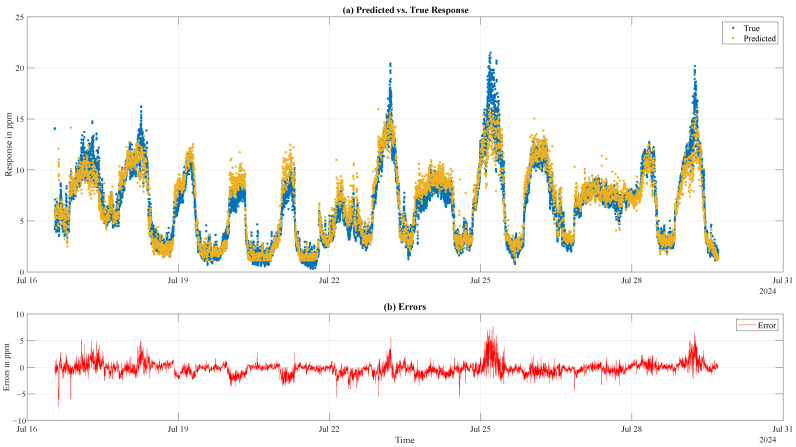
Predicted response of one BME688 equipped with a PM filter membrane (yellow dots) and the true interpolated concentration vales (blue dots) over a period of 13 days in location C (**a**). The errors are visualized in (**b**).

**Figure 16 sensors-25-02617-f016:**
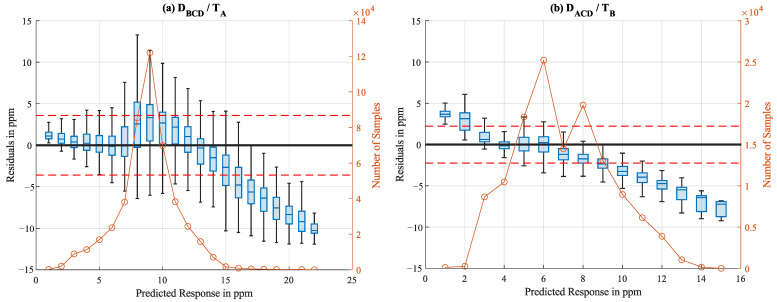
Boxplots of residuals from the Neural Networks trained with D_BCD_/tested with T_A_ (**a**) and D_ACD_/tested with T_B_ (**b**). The orange circles represent the number of test samples (interpolated data) for each box, while the dotted red line shows the RMSE.

**Table 1 sensors-25-02617-t001:** Comparison of three common types of ammonia gas sensing principles. Adapted from [10].

Parameter	Type of Gas Sensor		
	**MOS**	**Electrochemical**	**Infrared Absorption**
Sensitivity	++	+	++
Accuracy	+	+	++
Selectivity	-	+	++
Response time	++	-	-
Stability	+	- -	+
Durability	+	-	++
Maintenance	++	+	-
Cost	++	+	-
Suitability to portable instruments	++	-	- -

++: excellent, +: good, -: poor, - -: bad.

**Table 2 sensors-25-02617-t002:** Descriptions of the measurement locations in the pig farm.

Location	Description
A	Pregnant sows
B	6-week-old piglets
C	270-day-old sows
D	8-week-old piglets

**Table 3 sensors-25-02617-t003:** Evaluation of the design cycles Z_1_ and Z_2_.

Design Cycle	Temperature Cycle	Accuracy
**Z_1_**	HP354	80.89%
	HP301	89.90%
	HP411	81.67%
	HP501	87.31%
**Z_2_**	HP501	94.92%
	HP502	95.30%
	HP503	95.25%
	HP504	84.75%

**Table 4 sensors-25-02617-t004:** Results of the investigation of the sensor-to-sensor deviation and the impact of the filter membrane. The mean concentration of Location C used to normalize the RMSE values is given. In the upper part, the results of the cross validation for the evaluation of the senor-to-sensor deviation are shown with the (normalized) mean RMSE and the Std. Dev. for the two models. The lower part shows the NRMSE values for the trained models with data without a filter membrane (D_nF_) and with a filter membrane (D_F_).

Mean Concentration in Location C:	6.48 ppm			
	**Sensor-to-Sensor Deviation**			
	**Test Data**		**T_SD_**	
**Model**	**Mean RMSE ± Std. Dev. [ppm]**	**Mean NRMSE ± norm. Std. Dev. [%]**	**Mean RMSE ± Std. Dev. [ppm]**	**Mean NRMSE ± Norm. Std. Dev. [%]**
Neural Network	1.51 ± 0.04	23.3 ± 0.6	1.82 ± 0.24	28.1 ± 3.7
Bagged Trees	1.13 ± 0.01	17.4 ± 0.2	2.00 ± 0.21	30.9 ± 3.2
	**Filter Membrane Impact**			
**Training Dataset:**	**D_nF_**		**D_F_**	
**Model**	**RMSE [ppm]**	**NRMSE [%]**	**RMSE [ppm]**	**NRMSE [%]**
Neural Network	1.63	25.2	1.25	19.3
Bagged Trees	1.14	17.6	1.09	16.8

**Table 5 sensors-25-02617-t005:** Results of the investigation of the influence of different training scenarios. In each data split, the two trained models are tested with 25% test data and the data of one non-trained pig barn.

Mean Concentration:	5.51 ppm		10.45 ppm	
**Training Dataset: D_BCD_**	**Test Data**		**T_A_**	
**Model**	**RMSE [ppm]**	**NRMSE [%]**	**RMSE [ppm]**	**NRMSE [%]**
Neural Network	1.49	27.0	3.61	34.5
Bagged Trees	0.94	17.1	4.51	43.2
**Mean concentration:**	**7.44 ppm**		**4.62 ppm**	
**Training dataset: D_ACD_**	**Test data**		**T_B_**	
**Model**	**RMSE [ppm]**	**NRMSE [%]**	**RMSE [ppm]**	**NRMSE [%]**
Neural Network	1.76	23.7	2.21	47.8
Bagged Trees	1.21	16.3	2.09	45.2
**Mean concentration:**	**6.82 ppm**		**6.48 ppm**	
**Training dataset: D_ABD_**	**Test data**		**T_C_**	
**Model**	**RMSE [ppm]**	**NRMSE [%]**	**RMSE [ppm]**	**NRMSE [%]**
Neural Network	1.65	24.2	2.74	42.3
Bagged Trees	1.14	16.7	2.95	45.5
**Mean concentration:**	**7.18 ppm**		**5.40 ppm**	
**Training dataset: D_ABC_**	**Test data**		**T_D_**	
**Model**	**RMSE [ppm]**	**NRMSE [%]**	**RMSE [ppm]**	**NRMSE [%]**
Neural Network	2.08	29.0	2.58	47.8
Bagged Trees	1.41	19.6	2.42	44.8

## Data Availability

The Matlab scripts used for the analyses are available at https://github.com/MiguelBBecker/-Ammonia_Monitoring, accessed on 26 February 2025. The study data can be obtained upon request from Marion Gebhard (marion.gebhard@w-hs.de).

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
