# Peer review of "Virtual MOS Sensor Array Design for Ammonia Monitoring in Pig Barns"

_sensors, 2025, doi:10.3390/s25082617_

Round 1

Reviewer 1 Report

Comments and Suggestions for Authors

The paper provides in-dept experimental research of NH3 sensing using MOS sensors. The research investigates cyclic data acquisition, using advanced computing, and operating long period for assessment. The approach, procedure, and result highly contribute to the field of smart farm. It seems great job.

In abstract, author may provide quantifiable results.

In figure 3, the hardware has large volume that may contain many components. Thus, authors may address the details of the setup such as design, connection, and functions of each parts.

Author must revise the format according to journal's guideline there are minor errors like A,B,C,...,(a), (b),.. and (1), (2),... So, sentences, expressions, and symbols must be consistent.

Tables can be merged.

The author may provide a schematic of the sensor acquisition and processing system. In the future perspective, the author mentioned multi-sensing. It would be helpful for the reader to understand the possibilities if it provides information on how the sensor data is transmitted and which computer does the machine learning processing. A real pig farm considering smart farms and ammonia has thousands to tens of thousands of pigs. Can you suggest what approaches and technologies are needed for wider and multi-location sensing?

Reviewer 2 Report

Comments and Suggestions for Authors

The paper "Virtual MOS Sensor Array Design for Ammonia Monitoring in Pig Barns" presents a novel approach to ammonia detection using temperature-cycled Metal Oxide Semiconductor (MOS) sensors integrated with machine learning. The study aims to improve selectivity and accuracy by designing a virtual sensor array through an iterative data-driven approach. The sensors were deployed in four pig barns for two weeks, assessing sensor-to-sensor deviation, the impact of particulate matter (PM) filter membranes, and the transferability of the model across different environments. Results showed that PM filters improved sensor stability, but sensor-to-sensor variations and environmental differences significantly affected model performance. The study highlights the feasibility of using low-cost MOS sensors for continuous ammonia monitoring in livestock farming but emphasizes the need for further optimization, including cross-sensitivity mitigation and improved model generalization across diverse barn conditions. Here are some comments and questions to improve the manuscript:
1.  The study presents an innovative approach to designing a virtual MOS sensor array for ammonia detection in pig barns. However, it would benefit from a more explicit comparison with existing ammonia monitoring systems in terms of cost, performance, and practicality. Can the authors provide a direct comparison with other ammonia monitoring methods in a tabular form?
2.  The manuscript discusses sensor-to-sensor deviation and the impact of particulate matter (PM) filters. However, it does not address the long-term stability of MOS sensors in high-humidity, high-ammonia environments.
Introduction:
3. Discuss how their findings align with existing environmental regulations for livestock farms.
4. The introduction acknowledges that ammonia measurement faces interference from other gases. However, no details are provided on how the virtual sensor array differentiates ammonia from interfering compounds like methane (CH4) or hydrogen sulfide (H2S).
5. Improve your literature review by adding new references in E-nose such as DOI: 10.1016/j.snb.2022.131418.
Materials:
6. How frequently was the EC sensor calibrated, and was any drift correction applied?
6.  It is unclear whether the same environmental conditions were maintained across all barns.
7.  Was there a systematic approach, such as an optimization algorithm, to determine the best temperature profile?
Results:
8. The results show increased RMSE when testing on untrained sensors. Discuss whether post-processing or sensor normalization techniques could mitigate this variation.
9. Does the filter introduce any response time lag or reduce sensitivity to ammonia due to diffusion resistance? Experimental validation of this effect would be useful.
10.  Humidity levels in pig barns can fluctuate significantly. Have the authors analyzed how variations in humidity impact the MOS sensor resistance and whether compensation methods were applied?
Conclusion:
11. The authors propose optimizing the virtual sensor array with additional design cycles. Have they considered the integration of additional environmental sensors (e.g., humidity, temperature, or CO2) to improve ammonia quantification accuracy?

Reviewer 3 Report

Comments and Suggestions for Authors

This work aims to design a virtual sensor array for monitoring ammonia in pig barns. The main comments are given below:

  • Section 2 is very short. It should be adjusted. This section should provide a comprehensive state-of-the-art on the topic highlighting the knowledge gap of each paper. The authors should provide a table at the end of this section to compare previous research work and current work in the different aspects.
  • Section 3 provides information about the locations of the instaltions of different sensors (about 32 in this paper). It would be better to also add a diagram about the location inside the farm and the dimensions.
  • There is no information about the communication network part. How you collect data from the sensor nodes. What is the current topology and what technology is used for the communication network.
  • As shown in Figure 3, there are four sensor nodes. How data is collected from sensor nodes.
  • Please, review the captions of the figures. The current format of numbers 1, 2, 3 is not correct. For example, see Figs. 3, 12, 14, 15, etc.
  • The authors should add one section before the conclusion to explain the limitation of the current work and how to improve it.
  • The conclusion and future work could be improved, highlighting the contribution of the work and future research work. Extra information could move to the new section of limitation and implications of the work in section 5.
Comments on the Quality of English Language

 The English is fine.
